# Dirichlet-based Gaussian Processes for Large-scale Calibrated Classification

**Dimitrios Milios**
EURECOM
Sophia Antipolis, France
dimitrios.milios@eurecom.fr

**Raffaello Camoriano**
LCSL
IIT (Italy) & MIT (USA)
raffaello.camoriano@iit.it

**Pietro Michiardi**
EURECOM
Sophia Antipolis, France
pietro.michiardi@eurecom.fr

**Lorenzo Rosasco**
DIBRIS - Università degli Studi di Genova, Italy
LCSL - IIT (Italy) & MIT (USA)
lrosasco@mit.edu

**Maurizio Filippone**
EURECOM
Sophia Antipolis, France
maurizio.filippone@eurecom.fr

## Abstract

This paper studies the problem of deriving fast and accurate classification algorithms with uncertainty quantification. Gaussian process classification provides a principled approach, but the corresponding computational burden is hardly sustainable in large-scale problems and devising efficient alternatives is a challenge. In this work, we investigate if and how Gaussian process regression directly applied to classification labels can be used to tackle this question. While in this case training is remarkably faster, predictions need to be calibrated for classification and uncertainty estimation. To this aim, we propose a novel regression approach where the labels are obtained through the interpretation of classification labels as the coefficients of a degenerate Dirichlet distribution. Extensive experimental results show that the proposed approach provides essentially the same accuracy and uncertainty quantification as Gaussian process classification while requiring only a fraction of computational resources.

## 1 Introduction

Classification is a classic machine learning task. While the most basic performance measure is classification accuracy, in practice assigning a calibrated confidence to the predictions is often crucial [5]. For example in image classification, providing class predictions with a calibrated score is important to avoid making over-confident decisions [6, 12, 15]. Several classification algorithms that output a continuous score are not necessarily calibrated (e.g., support vector machines (SVMs) [24]). Popular ways to calibrate classifiers use a validation set to learn a transformation of their output score that recovers calibration; these include Platt scaling [24] and isotonic regression [39]. Calibration can also be achieved if a sensible loss function is employed [13], for example the logistic/cross-entropy loss, and it is known to be positively impacted if the classifier is well regularized [6].

Bayesian approaches provide a natural framework to tackle these kinds of questions, since quantification of uncertainty is of primary interest. In particular, Gaussian Processes Classification (GPC)

[8, 25, 36] combines the flexibility of Gaussian Processes (GPs) [25] and the regularization stemming from their probabilistic nature, with the use of the correct likelihood for classification, that is Bernoulli or multinomial for binary or multi-class classification, respectively. While we are not aware of empirical studies on the calibration properties of GPC, our results confirm the intuition that GPC is actually calibrated. The most severe drawback of GPC, however, is its computational burden, making it unattractive for large-scale problems.

In this paper, we study the question of whether GPs can be made efficient to find accurate and well-calibrated classification rules. A simple idea is to use GP regression directly on classification labels. This idea is quite common in non-probabilistic approaches [27, 34] and can be grounded from a decision theoretic point of view. Indeed, the Bayes' rule minimizing the expected least-squares is the expected conditional probability, which in classification is directly related to the conditional probabilities of each class (see e.g. [3, 31]). Performing regression directly on the labels leads to fast training and excellent classification accuracies [11, 17, 29]. However, the corresponding predictions are not calibrated for uncertainty quantification.

The main contribution of our work is the proposal of a transformation of the classification labels, which turns the original problem into a regression problem without compromising on calibration. For GPs, this has the enormous advantage of bypassing the need for expensive posterior approximations, leading to a method that is as fast as a simple regression carried out on the original labels. The proposed method is based on the interpretation of the labels as the output of a Dirichlet distribution, so we name it Dirichlet-based GP classification (GPD). Through an extensive experimental validation, including large-scale classification tasks, we demonstrate that GPD is calibrated and competitive in performance with state-of-the-art GPC[1] .

## 2   Related work

**Calibration of classifiers:**    Platt scaling [24] is a popular method to calibrate the output score of classifiers, as well as isotonic regression [39]. More recently, Beta calibration [13] and temperature scaling [6] have been proposed to extend the class of possible transformations and reduce the parameterization of the transformation, respectively. It is established that binary classifiers are calibrated when they employ the logistic loss; this is a direct consequence of the fact that the appropriate model for Bernoulli distributed variables is the one associated with this loss [13]. The extension to multi-class problems yields the so-called cross-entropy loss, which corresponds to the multinomial likelihood. Not necessarily, however, the right loss makes classifiers well calibrated; recent works on calibration of convolutional neural networks for image classification show that depth negatively impacts calibration due to the introduction of a large number of parameters to optimize, and that regularization is important to recover calibration [6].

**Kernel-based classification:**    Performing regression on classification labels is also known as least-squares classification [27, 34]. We are not aware of works that study GP-based least-squares classification in depth; we could only find a few comments on it in [25] (Sec. 6.5). GPC is usually approached assuming a latent process, which is given a GP prior, that is transformed into a probability of class labels through a suitable squashing function [25]. Due to the non-conjugacy between the GP prior and the non-Gaussian likelihood, applying standard Bayesian inference techniques in GPC leads to analytical intractabilities, and it is necessary to resort to approximations. Standard ways to approximate computations include the Laplace Approximation [36] and Expectation Propagation (EP, [19]); see, e.g., [16, 21] for a detailed review of these methods. More recently, there have been advancements in works that extend "sparse" GP approximations [35] to classification [9] in order to deal with the issues of scalability with the number of observations through the use of mini-batch-based optimization.

## 3   Background

Consider a multi-class classification problem. Given a set of $N$ training inputs $X = \{\mathbf{x}_1, \ldots, \mathbf{x}_N\}$ and their corresponding labels $Y = \{\mathbf{y}_1, \ldots, \mathbf{y}_N\}$, with one-hot encoded classes denoted by the vectors $\mathbf{y}_i$, a classifier produces a predicted label $\mathbf{f}(\mathbf{x}_*)$ as function of any new input $\mathbf{x}_*$.

In the literature, calibration is assessed through the *Expected Calibration Error* (ECE) [6], which is the average of the absolute difference between accuracy and confidence:

$$\text{ECE} = \sum_{m=1}^{M} \frac{|X_m|}{|X_*|} \left| \text{acc}(\mathbf{f}(X_m), Y_m) - \text{conf}(\mathbf{f}, X_m) \right|, \tag{1}$$

where the test set $X_*$ is divided into disjoint subsets $\{X_1, \ldots, X_M\}$, each corresponding to a given level of confidence $\text{conf}(\mathbf{f}, X_m)$ predicted by the classifier $\mathbf{f}$, while $\text{acc}(\mathbf{f}(X_m), Y_m)$ is the classification accuracy of $\mathbf{f}$ measured on the $m$-th subset. Other metrics used in this work to characterize the quality of a classifier are the *error rate* on the test set, and the *mean negative log-likelihood* (MNLL) of the test set under the classification model:

$$\text{MNLL} = -\frac{1}{|X_*|} \sum_{\mathbf{x}_*, \mathbf{y}_* \in X_*, Y_*} \log p(\mathbf{y}_* \mid X, Y, \mathbf{x}_*) \tag{2}$$

All metrics are defined so that lower values are better.

### 3.1 Kernel methods for classification

**GP classification (GPC)**   GP-based classification is defined by the following abstract steps:

1. A GP prior, which is characterized by mean function $\mu(\mathbf{x})$ and covariance function $k(\mathbf{x}, \mathbf{x}')$, is placed over a *latent* function $f(\mathbf{x})$. The GP prior is transformed by a sigmoid function so that the sample functions produce proper probability values. In the multi-class case, we consider $C$ independent priors over the vector of functions $\mathbf{f} = [f_1, \ldots, f_C]^\top$; transformation to proper probabilities is achieved by applying the softmax function $\boldsymbol{\sigma}(\mathbf{f})$ [2].

2. The observed labels $\mathbf{y}$ are associated with a categorical likelihood with probability components $p(y_c \mid \mathbf{f}) = \boldsymbol{\sigma}(\mathbf{f}(\mathbf{x}))_c$, for any $c \in \{1, \ldots, C\}$.

3. The latent posterior is obtained by means of Bayes' theorem.

4. The latent posterior is transformed via $\boldsymbol{\sigma}(\mathbf{f})$, to obtain a distribution over class probabilities.

Throughout this work, we consider $\mu(\mathbf{x}) = 0$ and covariance $k(\mathbf{x}, \mathbf{x}') = a^2 \exp\left(-\frac{(\mathbf{x} - \mathbf{x}')^2}{2l^2}\right)$, which is also known as the RBF kernel, and it is characterized by the $a^2$ and $l$ hyper-parameters, interpreted as the GP marginal variance and length-scale, respectively. The hyper-parameters are commonly selected my maximizing the marginal likelihood of the model.

The major computational challenge of GPC can be identified in Step 3 described above. The categorical likelihood implies that the posterior over the stochastic process is not Gaussian and it cannot be calculated analytically. Therefore, different approaches resort to different approximations of the posterior, for which we have $p(\mathbf{f} \mid X, Y) \propto p(\mathbf{f} \mid X) p(\mathbf{y} \mid \mathbf{f})$. For example in EP [19], local likelihoods are approximated by Gaussian terms so that the posterior has the following form:

$$p(\mathbf{f} \mid X, Y) \approx q(\mathbf{f} \mid X, Y) \propto p(\mathbf{f} \mid X) \mathcal{N}(\tilde{\boldsymbol{\mu}}, \tilde{\Sigma}) \tag{3}$$

where $\tilde{\boldsymbol{\mu}}$ and $\tilde{\Sigma}$ are determined by the *site parameters* learned through an iterative process. In variational classification approaches [9, 23], the approximating distribution $q(\mathbf{f})$ is directly parametrized by a set of *variational parameters*. Despite being successful, such approaches contribute significantly to the computational cost of GP classification, as they introduce a large number of parameters that need to be optimized. In this work, we explore a more straightforward Gaussian approximation to the likelihood that requires no significant computational overhead.

**GP regression (GPR) on classification labels**   A simple way to bypass the problem induced by categorical likelihoods is to perform least-squares regression on the labels by ignoring their discrete nature. This implies considering a Gaussian likelihood $p(\mathbf{y} \mid \mathbf{f}) = \mathcal{N}(\mathbf{f}, \sigma_n^2 I)$, where $\sigma_n^2$ is the observation noise variance. It is well-known that if the observed labels are 0 and 1, then the function $\mathbf{f}$ that minimizes the mean squared error converges to the true class probabilities in the limit of infinite data [26]. Nevertheless, by not squashing $\mathbf{f}$ through a softmax function, we can no longer guarantee that the resulting distribution of functions will lie within 0 and 1. For this reason, additional calibration steps are required (i.e. Platt scaling).

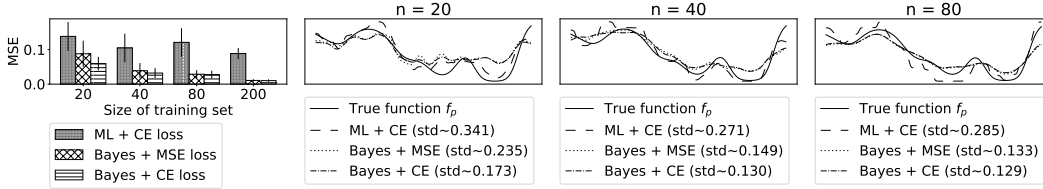

Figure 1: Convergence of classifiers with different loss functions and regularization properties. Left: summary of the mean squared error (MSE) from the true function $f_p$ for 1000 randomly sampled training sets of different size; the Bayesian CE-based classifier is characterized by smaller variance even when the number of training inputs is small. Right: demonstration of how the averaged classifiers approximate the true function for different training sizes.

**Kernel Ridge Regression (KRR) for classification**     The idea of performing regression directly on the labels is quite common when GP estimators are applied within a frequentist context [27]. Here they are typically derived from a non-probabilistic perspective based on empirical risk minimization, and the corresponding approach is dubbed Kernel Ridge Regression [7]. Taking this perspective, we make two observations. The first is that the noise and covariance parameters are viewed as regularization parameters that need to be tuned, typically by cross-validation. In our experiments, we compare this method with a canonical GPR approach. The second observation is that carrying out regression on the labels with least-squares can be justified from a decision-theoretic point of view. The Bayes' rule minimizing the expected least-squares is the regression function (the expected conditional probability), which in binary classification is proportional to the conditional probability of one of the two classes [3] (similar reasoning applies to multi-class classification [2, 20]). From this perspective, one could expect a least-squares estimator to be self-calibrated, however this is typically not the case in practice, a feature imputed to the limited number of points and the choice of function models. Post-hoc calibration has to be applied to both GPR- and KRR-based learning pipelines.

**Platt scaling**     Platt scaling [24] is an effective approach to perform post-hoc calibration for different types of classifiers, such as SVMs [22] and neural networks [6]. Given a decision function $f$, which is the result of a trained binary classifier, the class probabilities are given by the sigmoid transformation $\pi(\mathbf{x}) = \sigma(af(\mathbf{x}) + b)$, where $a$ and $b$ are optimised over a separate validation set, so that the resulting model best explains the data. Although this parametric form may seem restrictive, Platt scaling has been shown to be effective for a wide range of classifiers [22].

### 3.2   A note on calibration properties

We advocate that two components are critical for well-calibrated classifiers: *regularization* and the *cross-entropy loss*. Previous work indicates that regularization has a positive effect on calibration [6]. Also, classifiers that rely on the cross-entropy loss are reported to be well-calibrated [22]. This form of loss function is equivalent to the negative Bernoulli log-likelihood (or categorical in the multi-class case), which is the proper interpretation of classification outcomes.

In Figure 1, we demonstrate the effects of regularization and cross-entropy empirically: we summarize classification results on four synthetic datasets of increasing size. We assume that each class label is sampled from a Bernoulli distribution with probability given by the unknown function $f_p : \mathbb{R} \to [0, 1]$. For a classifier to be well-calibrated, it is sufficient that it accurately approximates $f_p$. We fit three kinds of classifiers: a maximum likelihood (ML) classifier that relies on cross entropy loss (CE), a Bayesian classifier with MSE loss (i.e. GPR classification), and finally a Bayesian classifier that relies on CE (i.e. GPC). We report the averages over 1000 iterations and the average standard deviations. The Bayesian classifiers that rely on the cross entropy loss converge to the true solution at a faster rate, and they are characterized by smaller variance.

Although performing GPR on the labels induces regularization through the prior, the likelihood model is not appropriate. One possible solution is to employ meticulous likelihood approximations such as EP or variational GP classification [9], alas at an often prohibitive computational cost, especially for considerably large datasets. In the section that follows, we introduce a methodology that combines the best of both worlds. We propose to perform GP regression on labels transformed in such a way that a less crude approximation of the categorical likelihood is achieved.

# 4   GP regression on transformed Dirichlet variables

There is an obvious defect in GP-based least-squares classification: each point is associated with a Gaussian likelihood, which is not the appropriate noise model for Bernoulli-distributed variables. Instead of approximating the true non-Gaussian likelihood, we propose to transform the labels in a latent space where a Gaussian approximation to the likelihood is more sensible.

For a given input, the goal of a Bayesian classifier is to estimate the distribution over its class probability vector; such a distribution is naturally represented by a Dirichlet-distributed random variable. More formally, in a $C$-class classification problem each observation $\mathbf{y}$ is a sample from a categorical distribution $\mathrm{Cat}(\boldsymbol{\pi})$. The objective is to infer the class probabilities $\boldsymbol{\pi} = [\pi_1, \ldots, \pi_C]^\top$, for which we use a Dirichlet model: $\boldsymbol{\pi} \sim \mathrm{Dir}(\boldsymbol{\alpha})$. In order to fully describe the distribution of class probabilities, we have to estimate the concentration parameters $\boldsymbol{\alpha} = [\alpha_1, \ldots, \alpha_C]^\top$. Given an observation $\mathbf{y}$ such that $y_k = 1$, our best guess for the values of $\boldsymbol{\alpha}$ will be: $\alpha_k = 1 + \alpha_\epsilon$ and $\alpha_i = \alpha_\epsilon, \forall i \neq k$. Note that it is necessary to add a small quantity $0 < \alpha_\epsilon \ll 1$, so as to have valid parameters for the Dirichlet distribution. Intuitively, we implicitly induce a Dirichlet prior so that before observing a data point we have the probability mass shared equally across $C$ classes; we know that we should observe exactly one count for a particular class, but we do not know which one. Most of the mass is concentrated on the corresponding class when $\mathbf{y}$ is observed. This practice can be thought of as the categorical/Bernoulli analogue of the noisy observations in GP regression. The likelihood model is:

$$p(\mathbf{y} \mid \boldsymbol{\alpha}) = \mathrm{Cat}(\boldsymbol{\pi}), \quad \text{where } \boldsymbol{\pi} \sim \mathrm{Dir}(\boldsymbol{\alpha}). \tag{4}$$

It is well-known that a Dirichlet sample can be generated by sampling from $C$ independent Gamma-distributed random variables with shape parameters $\alpha_i$ and rate $\lambda = 1$; realizations of the class probabilities can be generated as follows:

$$\pi_i = \frac{x_i}{\sum_{c=1}^C x_c}, \quad \text{where } x_i \sim \mathrm{Gamma}(\alpha_i, 1) \tag{5}$$

Therefore, the noisy Dirichlet likelihood assumed for each observation translates to $C$ independent Gamma likelihoods with shape parameters either $\alpha_i = 1 + \alpha_\epsilon$, if $y_i = 1$, or $\alpha_i = \alpha_\epsilon$ otherwise.

In order to construct a Gaussian likelihood in the log-space, we approximate each Gamma-distributed $x_i$ with $\tilde{x}_i \sim \mathrm{Lognormal}(\tilde{y}_i, \tilde{\sigma}_i^2)$ through moment matching (mean and variance):

$$\mathrm{E}[x_i] = \mathrm{E}[\tilde{x}_i] \Leftrightarrow \alpha_i = \exp(\tilde{y}_i + \tilde{\sigma}_i^2/2)$$
$$\mathrm{Var}[x_i] = \mathrm{Var}[\tilde{x}_i] \Leftrightarrow \alpha_i = \left(\exp(\tilde{\sigma}_i^2) - 1\right)\exp(2\tilde{y}_i + \tilde{\sigma}_i^2)$$

Thus, for the parameters of the normally distributed logarithm we have:

$$\tilde{y}_i = \log \alpha_i - \tilde{\sigma}_i^2/2, \qquad \tilde{\sigma}_i^2 = \log(1/\alpha_i + 1) \tag{6}$$

Note that this is the first approximation to the likelihood that we have employed so far. One could argue that a log-Normal approximation to a Gamma-distributed variable is reasonable, although it is not accurate for small values of the shape parameter $\alpha_i$. However, the most important implication is that we can now consider a Gaussian likelihood in the log-space. Assuming a vector of latent processes $\mathbf{f} = [f_1, \ldots, f_C]^\top$, we have:

$$p(\tilde{y}_i \mid \mathbf{f}) = \mathcal{N}(f_i, \tilde{\sigma}_i^2), \tag{7}$$

where class labels in the transformed logarithmic space are now denoted by $\tilde{y}_i$. We note that each observation is associated with a different noise parameter $\tilde{\sigma}_i^2$, yielding a *heteroskedastic* regression model. In fact, the $\tilde{\sigma}_i^2$ values (as well as $\tilde{y}_i$) solely depend on the Dirichlet pseudo-count assumed in the prior, which has only two possible values. Given this likelihood approximation, it is straightforward to place a GP prior over $\mathbf{f}$ and evaluate the posterior over the $C$ latent processes exactly. The multivariate GP prior does not assume any prior covariance across classes, meaning that they are assumed to be independent a priori. It is possible to make kernel parameters independent across processes, or shared so that they are informed by all classes.

**Remark:** In the binary classification case, we still have to perform regression on two latent processes. The use of heteroskedastic noise model implies that one latent process is not a mirrored version of the other (see Figure 2), contrary to GPC.

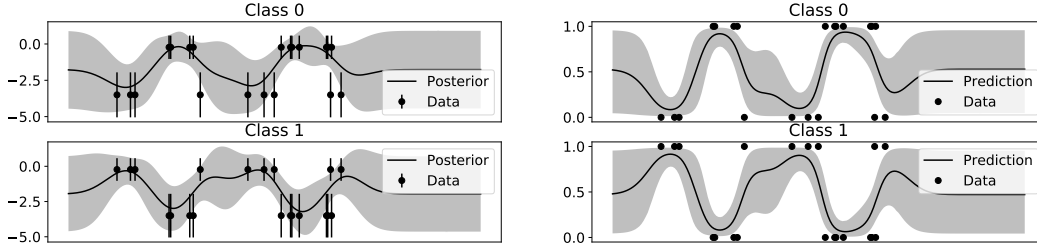

Figure 2: Example of Dirichlet regression for a one-dimensional binary classification problem. Left: the latent GP posterior for class "0" (top) and class "1" (bottom). Right: the transformed posterior through softmax for class "0" (top) and class "1" (bottom).

### 4.1 From GP posterior to Dirichlet variables

The obtained GP posterior emulates the logarithm of a stochastic process with Gamma marginals that gives rise to the Dirichlet posterior over class labels. It is straightforward to sample from the posterior log-Normal marginals, which should behave approximately as Gamma-distributed samples to generate posterior Dirichlet samples as in Equation (5), which corresponds to a simple application of the softmax function on the samples from the GP posterior. The expectation of class probabilities is:

$$\mathrm{E}[\pi_{i,*} \mid X, Y, \mathbf{x}_*] = \int \frac{\exp(f_{i,*})}{\sum_j \exp(f_{j,*})} \, p(f_{i,*} \mid X, Y, \mathbf{x}_*) \, d\mathbf{f}_* \,, \tag{8}$$

which can be approximated by sampling from the Gaussian posterior $p(f_{i,*} \mid X, Y, \mathbf{x}_*)$.

Figure 2 shows an example of Dirichlet regression for a one-dimensional binary classification problem. The left panels demonstrate how the GP posterior approximates the transformed data; the error bars represent the standard deviation for each data-point. Notice that the posterior for class "0" (top) is not a mirror image of class "1" (bottom), because of the different noise terms assumed for each latent process. The right panels show results in the original output space, after applying softmax transformation; as expected in the binary case, one posterior process is a mirror image of the other.

### 4.2 Optimizing the Dirichlet prior $\alpha_\epsilon$

The performance of Dirichlet-based classification is affected by the choice of $\alpha_\epsilon$, in addition to the usual GP hyper-parameters. As $\alpha_\epsilon$ approaches zero, $\alpha_i$ converges to either $1$ or $0$. It is easy to see that for the transformed "1" labels we have $\tilde{\sigma}_i^2 = \log(2)$ and $\tilde{y}_i = \log(1/\sqrt{2})$ in the limit. The transformed "0" labels, however, converge to infinity, and so do their variances. The role of $\alpha_\epsilon$ is to make the transformed labels finite, so that it is possible to perform regression. The smaller $\alpha_\epsilon$ is, the further the transformed labels will be apart, but at the same time, the variance for the "0" label will be larger.

By increasing $\alpha_\epsilon$, the transformed labels of different classes tend to be closer. The marginal log-likelihood tends to be larger, as it is easier for a zero-mean GP prior to fit the data. However, this behavior is not desirable for classification purposes. For this reason, the Gaussian marginal log-likelihood in the transformed space is not appropriate to determine the optimal value for $\alpha_\epsilon$.

Figure 3 demonstrates the effect of $\alpha_\epsilon$ on classification accuracy, as reflected by the MNLL metric. Each subfigure corresponds to a different dataset; MNLL is reported for different choices of $\alpha_\epsilon$ between $0.1$ and $0.001$. As a general remark, it appears that there is no globally optimal $\alpha_\epsilon$ parameter across datasets. However, the reported training and test MNLL curves appear to be in agreement regarding the optimal choice for $\alpha_\epsilon$. We therefore propose to select the $\alpha_\epsilon$ value that minimizes the MNLL on the training data.

## 5 Experiments

We experimentally evaluate the methodologies discussed on the datasets outlined in Table 1. For the implementation of GP-based models, we use and extend the algorithms available in the GPflow

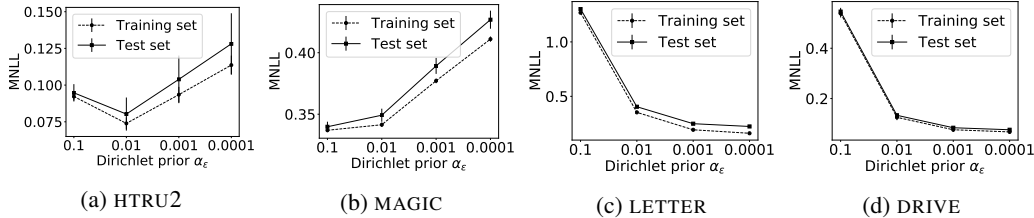

|        (a) HTRU2        |        (b) MAGIC        |        (c) LETTER        |        (d) DRIVE        |

Figure 3: Exploration of $\alpha_\epsilon$ for 4 different datasets with respect to the MNLL metric.

Table 1: Datasets used for evaluation, available from the UCI repository [1].

| Dataset | Classes | Training instances | Test instances | Dimensionality | Inducing points |
|---|---|---|---|---|---|
| EEG | 2 | 10980 | 4000 | 14 | 200 |
| HTRU2 | 2 | 12898 | 5000 | 8 | 200 |
| MAGIC | 2 | 14020 | 5000 | 10 | 200 |
| MINIBOO | 2 | 120064 | 10000 | 50 | 400 |
| COVERBIN | 2 | 522910 | 58102 | 54 | 500 |
| SUSY | 2 | 4000000 | 1000000 | 18 | 200 |
| LETTER | 26 | 15000 | 5000 | 16 | 200 |
| DRIVE | 11 | 48509 | 10000 | 48 | 500 |
| MOCAP | 5 | 68095 | 10000 | 37 | 500 |

library [18]. More specifically, for GPC we make use of variational sparse GP [8], while for GPR we employ sparse variational GP regression [35]. The latter is also the basis for our GPD implementation: we apply adjustments so that heteroskedastic noise is admitted, as dictated by the Dirichlet mapping. Concerning KRR, in order to scale it up to large-scale problems we use a subsampling-based variant named Nyström KRR (NKRR) [33, 37]. Nyström-based approaches have been shown to achieve state-of-the-art accuracy on large-scale learning problems [4, 14, 28, 30, 32]. The number of inducing (subsampled) points used for each dataset is reported in Table 1.

The experiments have been repeated for 10 random training/test splits. For each iteration, inducing points are chosen by applying k-means clustering on the training inputs. Exceptions are COVERBIN and SUSY, for which we used 5 splits and inducing points chosen uniformly at random. For GPR we further split each training dataset: 80% of which is used to train the model and the remaining 20% is used for calibration with Platt scaling. NKRR uses an 80-20% split for $k$-fold cross-validation and Platt scaling calibration, respectively. For each of the datasets, the $\alpha_\epsilon$ parameter of GPD was selected according to the training MNLL: we have 0.1 for COVERBIN, 0.001 for LETTER, DRIVE and MOCAP, and 0.01 for the remaining datasets.

In all experiments, we consider an isotropic RBF kernel; the kernel hyper-parameters are selected by maximizing the marginal likelihood for the GP-based approaches, and by $k$-fold cross validation for NKRR (with $k = 10$ for all datasets except from SUSY, for which $k = 5$). In the case of GPD, kernel parameters are shared across classes so they are informed by all classes. In the case of GPR, we also optimize the noise variance jointly with all kernel parameters.

The performance of GPD, GPC, GPR and NKRR is compared in terms of various error metrics, including error rate, MNLL and ECE for a collection of datasets. The obtained error rate, MNLL and ECE values are summarized in Figure 4. The GPC method tends to outperform GPR in most cases. Regarding the GPD approach, its performance tends to lie between GPC and GPR; in some instances classification performance is better than GPC and NKRR. Most importantly, this performance is obtained at a fraction of the computational time required by the GPC method. Figure 5 summarizes the speed-up achieved during hyper-parameter optimization by GPD in comparison with the variational GP classification approach. In the context of sparse variational regression, these computational gains are a consequence of closed-form results for the optimal variational distribution [35], which are not available for non-Gaussian likelihoods. We note that hyper-parameter and variational optimization have been performed using the ScipyOptimizer class of GPflow, which applies early stopping if convergence is detected. Convergence for GPD is faster simply because optimization involves fewer parameters. A more detailed exposition can be found in the supplementary material.

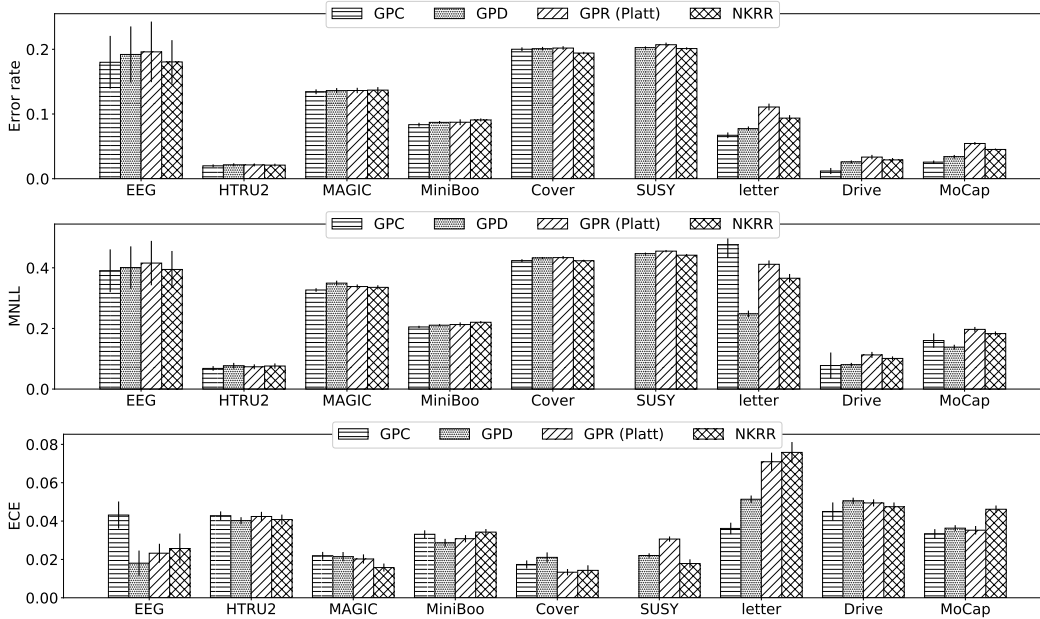

Figure 4: Error rate, MNLL and ECE for the datasets considered in this work.

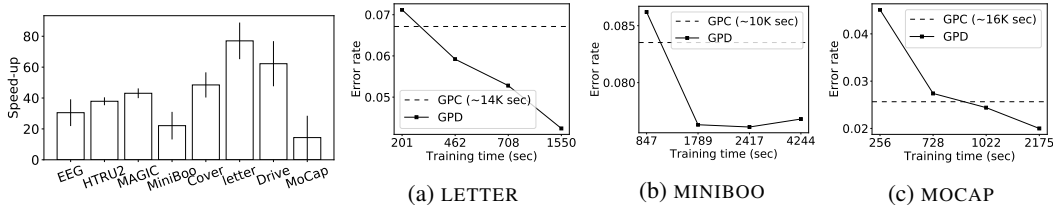

(a) LETTER        (b) MINIBOO        (c) MOCAP

Figure 5: Left: Speed-up obtained by using GPD as opposed to GPC. Right: Error vs training time for GPD as the number of inducing points is increased for three datasets. The dashed line represents the error obtained by GPC using the same number of inducing points as the fastest GPD listed.

This dramatic difference in computational efficiency has some interesting implications regarding the applicability of GP-based classification methods on large datasets. GP-based machine learning approaches are known to be computationally expensive; their practical application on large datasets demands the use of scalable methods to perform approximate inference. The approximation quality of sparse approaches depends on the number (and the selection) of inducing points. In the case of classification, the speed-up obtained by GPD implies that the saved computational budget can be spent on a more fine-grained sparse GP approximation. In Figure 5, we explore the effect of increasing the number of inducing points $N_u$ for three datasets: LETTER with $N_u \in \{500, 800, 1000, 1600\}$, MINIBOO with $N_u \in \{400, 500, 600, 800\}$ and MOCAP with $N_u \in \{500, 800, 1000, 1600\}$. Regarding GPC, we fix the computational budget to the smallest $N_u$ in each case. We see that the error rate for GPD drops significantly as the budget is increased; however, the latter remains a fraction of the original GPC computational effort.

Finally, we acknowledge that the computational cost of variational GPC can be reduced by means of mini-batches-based training [8, 10, 38]. In the supplementary material, we perform a detailed comparison between GPD and variational GPC with mini-batches [8]. The efficiency of GPC with a carefully selected mini-batch size is significantly improved, although stochastic optimization is characterized by slower convergence compared to full-batch-based optimization. As a result, GPD convergence remains faster for most datasets. This advantage becomes more obvious in scenarios where hyper-parameters are either known or reused, since no optimization step is required for a regression-based method.

# 6 Conclusions

Most GP-based approaches to classification in the literature are characterized by a meticulous approximation of the likelihood. In this work, we experimentally show that such GP classifiers tend to be well-calibrated, meaning that they correctly estimate classification uncertainty, as this is expressed through class probabilities. Despite this desirable property, their applicability is limited to small/moderate size of datasets, due to the high computational complexity of approximating the true posterior distribution.

Least-squares classification, which may be implemented either as GPR or KRR, is an established practice for more scalable classification. However, the crude approximation of a non-Gaussian likelihood with a Gaussian one has a negative impact on classification quality, especially as this is reflected by the calibration properties of the classifier.

Considering the strengths and practical limitations of GPs, we proposed a classification approach that is essentially an heteroskedastic GP regression on a latent space induced by a transformation of the labels, which are viewed as Dirichlet-distributed random variables. This allowed us to convert $C$-class classification to a problem of regression involving $C$ latent processes with Gamma likelihoods. We then proposed to approximate the Gamma-distributed variables with log-Normal ones, and thus we achieved a sensible Gaussian approximation in the logarithmic space. Crucially, this can be seen as a pre-processing step, that does not have to be learned, unlike in GPC, where an accurate transformation is sought iteratively. Our experimental analysis shows that Dirichlet-based GP classification produces well-calibrated classifiers without the need for post-hoc calibration steps. The performance of our approach in terms of classification accuracy tends to lie between properly-approximated GPC and least-squares classification, but most importantly it is orders of magnitude faster than GPC.

As a final remark, we note that the predictive distribution of the GPD approach is different from that obtained by GPC, as can be seen in the extended results in the supplementary material. An extended characterization of the predictive distribution for GPD is subject of future work.

### Acknowledgments

L. R. acknowledges the financial support of the Center for Brains, Minds and Machines (CBMM), funded by NSF STC award CCF-1231216, the Italian Institute of Technology, the AFOSR projects FA9550-17-1-0390 and BAA-AFRL-AFOSR-2016-0007 (European Office of Aerospace Research and Development), and the EU H2020-MSCA-RISE project NoMADS - DLV-777826. R. C. and L. R. gratefully acknowledge the support of NVIDIA Corporation for the donation of the Titan Xp GPUs and the Tesla k40 GPU used for this research. DM and PM are partially supported by KPMG. MF gratefully acknowledges support from the AXA Research Fund.

## Footnotes

[1] The code is available at `https://github.com/dmilios/dirichletGPC`.

[2]Softmax function $\boldsymbol{\sigma}(\mathbf{f})$ s.t. $\sigma(\mathbf{f})_j = \exp(f_j) / \sum_{c=1}^{C} \exp(f_c)$ for $j = 1, \ldots C$

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
