[Supplementary Material]

# Dirichlet-based Gaussian Processes
# for Large-scale Calibrated Classification
# Supplementary Material

**Dimitrios Milios**
EURECOM
Sophia Antipolis, France
dimitrios.milios@eurecom.fr

**Raffaello Camoriano**
LCSL
IIT (Italy) & MIT (USA)
raffaello.camoriano@iit.it

**Pietro Michiardi**
EURECOM
Sophia Antipolis, France
pietro.michiardi@eurecom.fr

**Lorenzo Rosasco**
DIBRIS - Università degli Studi di Genova, Italy
LCSL - IIT (Italy) & MIT (USA)
lrosasco@mit.edu

**Maurizio Filippone**
EURECOM
Sophia Antipolis, France
maurizio.filippone@eurecom.fr

## 1 Exploration of Dirichlet prior $\alpha_\epsilon$

In Section 4 we have introduced $\alpha_\epsilon$, which is a noise term that corrupts the labels, so that any one-hot encoded vector $\mathbf{y}$ can be mapped to a valid Dirichlet distribution. The vector with values equal to $\alpha_\epsilon$ can be interpreted as the parameters of a Dirichlet distribution prior to the observation of a class label. Then, the corrupted label will be given as the parameter vector of the posterior Dirichlet distribution:

$$\boldsymbol{\alpha} = \mathbf{y} + \alpha_\epsilon$$

As we have established in Section 4.2, the choice of $\alpha_\epsilon$ may have a significant effect on the performance of Dirichlet-based classification. We have also argued that the GP marginal log-likelihood in the transformed space is not an appropriate measure to determine $\alpha_\epsilon$. In our experience, however, it is possible to identify a reasonably good value for $\alpha_\epsilon$ by observing its effect on the MNLL metric as measured on the training data. Figure 1 is a summary of our experimentation on all datasets with $\alpha_\epsilon \in \{0.1, 0.01, 0.001, 0.0001\}$. Although there is no consensus regarding the optimal value for $\alpha_\epsilon$ across datasets, we see that in most cases the training value for MNLL reflects the corresponding value reported for the test set.

## 2 Extended calibration results

*Reliability diagrams* offer a visual representation of calibration properties, where accuracy is plotted as a function of confidence for the subsets $\{X_1, \ldots, X_M\}$. For a perfectly calibrated classifier, the accuracy function should be equal to the identity line, implying that $\mathrm{accu}(X_m) = \mathrm{conf}(X_m)$. Large deviations from the identity line mean that the class probabilities are either underestimated or overestimated.

In Figure 2 we summarize the reliability diagrams for a number of binary classification datasets. Each row of diagrams corresponds to a particular dataset; and each column to one of the GP-based

Figure 1: Exploration of $\alpha_\epsilon$ with respect to the MNLL metric.

classification approaches that we have discussed in this work. In particular, we consider the variational GP classification algorithm of [2] (GPC), our Dirichlet-based classification scheme (GPD), and GP regression on the labels without and with a Platt-scaling post-hoc calibration step (GPR and GPR (PLATT)). Note that each one of these approaches produces a distribution of classifiers. Thus, in the diagrams of Figure 2 we show the reliability curve of the mean classifier (depicted as solid lines-points), along with the classifiers described by the upper and lower 95% quantiles of the predictive distribution (grey area). If a classifier is well-calibrated, then its reliability curve should be close to the identity curve (dashed line); the latter should also lie within the limits of the grey area.

For the results of Figure 2, we have considered $M = 10$ subsets for different levels of confidence. We note that deviations from the identity curve should not be deemed important, if these are not backed by a sufficient number of samples. For some datasets there are certain levels of confidence (middle section of HTRU2 for example) that contain very few data. In order to reflect this behavior, we also plot the histograms showing the proportion of the test set that corresponds to each confidence level.

A careful inspection of Figure 2 suggests that both GPC and GPD produce well-calibrated models. The GPR approach, on the other hand, tends to produce a sigmoid-shaped reliability curve, which suggests that there is underestimation of the class probabilities. This behavior is cured however by performing calibration via Platt-scaling, as we see for the GPR (PLATT) method.

These conclusions are further supported by Figure 3, which summarizes the reliability plots for a number of multi-class datasets. In the multi-class case, it is not obvious how to concisely summarise the effect of the predictive distribution, so we resort to simple reliability plots of the average classifier for each method.

As a final remark, we note that the predictive distribution of the GPD approach is different from that obtained by GPC. In fact, judging form the upper and lower quantile classifiers as presented in Figure 2, it appears that GPD results in a narrower predictive distribution, which is nevertheless

well-calibrated. An extended characterization of the predictive distribution for GPD is subject of future work.

Figure 2: Reliability diagrams for four different GP-based classification approaches on four binary classification datasets. The bounds correspond to the classifiers given by the 95% confidence interval of the posterior GP.

## 3 Discussion on computational efficiency

The speed-up values reported in Figure 5 of the main paper correspond to the difference between GPD and GPC regarding the execution time required for hyperparameter optimization. For each comparison, we use an identical setup (i.e. number or cores, memory) in the Zoe Analytics cluster [5]. In this section, we provide some additional insights on the nature of the speed-up achieved.

If we assume fixed values for the GP hyperparameters, then the difference in computational efficiency between GPD and GPC becomes obvious. For example, expectation propagation [4] involves optimising a large number of *site parameters* that control the local Gaussian approximations to the likelihood; this is achieved by an iterative process that requires carrying out several matrix factorizations. Similarly, variational GPC approaches, such as [1], require optimising the *variational parameters*. In contrast with these classical GPC approaches, likelihood approximation in GPD is efficiently accomplished by means of moment matching, similar to least-squares classification.

Figure 3: Reliability diagrams for three multiclass classification datasets.

| (a) EEG | (b) HTRU2 | (c) MAGIC | (d) MINIBOO |

| (e) COVERBIN | (f) LETTER | (g) DRIVE | (h) MOCAP |

Figure 4: Progression of MNLL for the test set for GPC with different mini-batch sizes and GPD. All methods have been optimized with the same number of iterations/epochs.

In case hyperparameter optimisation is performed, any expensive likelihood approximations will have to be carried out many times as the hyperparameter space is explored. Some recent works [1, 3, 6] propose methodologies that involve a joint optimisation of the hyperparameters along with any parameters for likelihood approximations. Although such a practice has significant implications on the performance, our approach remains more computationally efficient in comparison, since the optimisation step simply involves fewer parameters, i.e. the GP hyperparameters only, as opposed to a large number of variational parameters. In fact, the speed-ups that have been reported in Figure 5 of Section 5 reflect the performance difference between GPD and the variational sparse GP of [1]. A deeper insight into the source of this difference is provided by Figure 4, which illustrates the progression of MNLL for a number of datasets, as this is measured on the test set, during the course of the optimisation process for both GPD and variational GPC. We can see that MNLL approaches its minimal value in a significantly smaller number of iterations compared to GPC. The running time for a single iteration is virtually identical for both approaches, but the difference in total running time until convergence is dramatic, as can be seen in Figure 4.

Finally, we acknowledge that computational efficiency of variational GPC can be improved by admitting stochastic gradient descent optimization realised by means of mini-batches [1]. A full pass over the training set, also called an *epoch*, involves several gradient updates, which are significantly less expensive than a full GPD or GPC iteration. The effects of this practice can be seen in Figure 4, where we report the MNLL progression for GPC with different mini-batch sizes over the course of epochs. For all datasets, the use of mini-batches has a dramatic impact on performance compared to the full GPC, as this is reflected by the progression of MNLL. In most cases, the test MNLL value converges significantly faster to its lowest value compared to full-batch GPC, and in some cases it approaches the performance of our approach. Nevertheless, GPD convergence remains faster for most datasets, especially in the early stages of the optimization process. This advantage of a regression-based approach becomes more evident in cases where the hyperparameters are easily identifiable or reusable, as no optimisation is required.