[Reviews · NeurIPS 2018]

Reviewer 1



The authors of this paper introduce a novel approach to GP classification, called GPD. The authors use a GP to produce the parameters of a Dirichlet distribution, and use a categorical likelihood for multi-class classification problems. After applying a log-normal approximation to the Dirichlet distribution, inference for GPD is the same as exact-GP inference (i.e. does not require EP, Laplace approximation, etc.) The authors show that GPD has competitive accuracy, is well calibrated, and offers a speedup over existing GP-classificaiton methods. Quality: The method introduced by this paper is a clever probabilistic formulation of Bayesian classification. The method is well-grounded in Bayesian statistics and makes good comparisons to similar existing approaches. The one issue I have with this paper are the claims about the speedup'' afforded by this method (see detailed comments). The authors are not specific about the speedup (is it training time or inference time?) and I'm concerned they are missing an important baseline. Clarity: The paper is clearly written. Originality: To the best of my knowledge this is a novel approach to Bayesian classification. Significance: The idea is an interesting new approach to approximate inference for classification, though it does not perform better than other classification methods (in terms of accuracy). Nevertheless, it will be of interest to the Bayesian ML community. Overall: The proposed method is a novel approach to classification - and based on these merits I would be willing to accept the paper. However, the authors need to substantiate the speedup results with more detail and analysis if they wish to report this as a primary advantage. Detailed comments: My main concern with this paper is with regards to the speedup'' that the authors claim. There are many missing details. The authors claim that their GPD method will be faster than GP classification because "GPC requires carrying out several matrix factorizations." (What matrix factorizations are the authors referring to?) The inducing-point GP-classification method in the results (SVGP) is asymptotically the same as the inducing-point method used for GPD (SGPR). The NMLL computations (used for training) and inference computations for SVGP and SGPR both require the Cholesky decomposition of an m x m matrix (where m is the number of inducing points). This is the primary computational bottleneck of both SVGP (for GP classification) and SGPR (for GPD). I'm assuming the authors claim that GPD is faster because GPD does not require learning variational parameters. The training'' of GPD - as far as I understand - only requires optimizing the hyperparameters, which presumably requires fewer optimization iterations. However, one key advantage of SVGP (for GP classification) is that it can be optimized with SGD. So even though SVGP requires more optimization iterations, each iteration will be fast since it only requires a minibatch of data. On the other hand, every optimization iteration of GPD will require the entire dataset. Therefore, there is not an obvious reason why GPD (and GP regression) should be that much faster than GP classification. The authors should elaborate with more detail in Sections 2 and 3 as to why GP-classification is slow. While the authors do report some empirical speedups, the experiments are missing several important details. Firstly, what is the speedup'' measuring - training or inference? (I'm assuming that it is measuring training, but this must be clarified.) Secondly, was the author's GP-classification baseline trained with or without SGD? If it was trained without SGD, I would argue that the speed comparisons are unfair since stochastic optimization is key to making GP-classification with SVGP a fast method. Small comments: - 149: "For a classifier to be well-calibrated, it should accurately approximate fp." - This is not necessarily true. Assuming a balanced dataset without loss of generality, a latent function that outputs zero (in the case of CE) or 0.5 (in the case of least-squares classification) for all samples is well calibrated. - Figure 1: it is difficult to differentiate between the lines in the n=60 and n=80 plots - 241: do you mean the initial inducing points were chosen by k-means? SVGP and SGPR should optimize the inducing point locations. ----------------------- Post author response: While I would still vote to accept the paper, I will note that the author's response did not clarify my questions about the speedup. (Was it training or inference?) Additionally, the author's claim about speed still does not acknowledge many recent advances in scalable GP classification: "Our claim in the paper about multiple matrix factorizations pertains to training times and it is specific to the case of standard implementations of EP and the Laplace approximation." Though the authors make a statement about GPC speed based on the EP/Laplace approximations, yet they do not compare to these methods in their experiments! Regarding SGD-based GPC methods: there have been a number of recent works in scalable GPC methods that rely on SGD [1-3]. These papers report large empirical speedups when using SGD-based optimization over batch GD optimization. By not comparing against SGD-based methods, the authors are disregarding GPC advances from the past 4 years. That being said, I still do believe that this method is novel, and I would vote for acceptance. I would encourage the authors to be much more rigorous about their speedup claims. There are some scenarios where a non-GPC-based approach would be clearly advantageous (e.g. in an active learning setting, when the kernel hyperparameters can be reused between data acquisition), and I would suggest the authors to focus on these specific scenarios rather than making broad claims about speed. [1] Hensman, James, Alexander Matthews, and Zoubin Ghahramani. "Scalable Variational Gaussian Process Classification." Artificial Intelligence and Statistics. 2015. [2] Hernández-Lobato, Daniel, and José Miguel Hernández-Lobato. "Scalable Gaussian process classification via expectation propagation." Artificial Intelligence and Statistics. 2016. [3] Wilson, Andrew G., et al. "Stochastic variational deep kernel learning." Advances in Neural Information Processing Systems. 2016.

Reviewer 2



The authors propose a novel approximation for multi-class Gaussian process classification. The proposed approximation allows GP classification problems to be solved as GP regression problems and thus yielding a significant speed up. The core idea is to represent the likelihood as a categorical distribution with a Dirichlet prior on the parameters and then use the fact that a Dirichlet distribution can be represented using a set of independent gamma distributions. These independent gamma distributions are then approximated using log-normal distributions via moment matching. The result is a model with heteroscedastic Gaussian likelihood. The downside is that the approximation introduces a new parameter that cannot be tuned using marginal likelihood. Besides the model approximation introduced in the paper, the authors also rely on standard approximate inference methods to make their method scale, i.e. variational inference and inducing points. The proposed method is evaluated and compared to benchmark methods using 9 different datasets. The experiments shows that the performance of the proposed method is comparable to the reference methods in terms of classification error, mean-negative log likelihood and expected calibration error. The paper appears technically sound and correct. The author do not provide any theoretical analysis for the proposed method, but the claims for the method are supported by experimental results for 9 datasets showing that the method is comparable to the reference methods. My only concern is the parameter alpha_eps introduced in the model approximation. In figure 3, the authors show that the performance is quite sensitive to the value of alpha_eps. However, they also show that optimal value for alpha_eps (in terms of MNLL) is strongly correlated between the training and test set. However, they only show this for 4 out of the 9 data sets included in the paper. It would be more convincing if the authors would include the corresponding figures for the remaining 5 datasets in the supplementary material along with the reliability diagrams. The paper is in general clear, well-organized, and well-written. However, there are a couple details that could be made more clear. If I understand the method correctly, then solving a classification problem with C classes essentially boils down to solving C independent GP regression problems each with different likelihoods, where each regression problem is basically solving a one-vs-rest classification problem. After solving the C regression problems, the posterior class probabilities can be computed by sampling from C GP posteriors using eq. (7). 1) If this is correct, the authors should mention explicitly that each regression problem is solved independently and discuss the implications. 2) How are the hyperparameters handled? Is each regression problem allowed to have different hyperparameters or do they share the same kernel? 3) In my opinion, the authors should also state the full probabilistic model before and after the approximation. That would also help clarifying point 1). The idea of solving classification problems as regression problem is not new. However, to the best of my knowledge, the idea of representing the multi-class classification likelihood in terms of independent gamma distributions and then approximating each gamma distribution using a lognormal distribution is indeed original and novel. The authors propose a method for making GP classification faster. I believe that these results are important and that both researchers and practitioners will have an interest in using and improving this method. Further comments Line 89: “The observable (non-Gaussian) prior is obtained by transforming”. What is meant by an observable prior? Line 258: “Figure 5 summarizes the speed-up achieved by the used of GPD as opposed to the variational GP classification approach“. 1) Typo, 2) For completeness, the authors should add a comment on where the actual speed up comes from. I assume it’s due to the availability of closed-form results for the optimal variational distribution in (Titsias, 2009) which are not available for non-Gaussian likelihoods? Line 266: “In Figure 5, we explore the effect of increasing the number of inducing points for three datasets” Can you please elaborate on the details? --------------------------------------------------------------------- After author feedback --------------------------------------------------------------------- I've read the rebuttal and the authors addressed all my concerns properly. Therefore, I have updated my overall score from 6 to 7.

Reviewer 3



This paper is concerned with multi-class classification using GPs, placing particular emphasis on producing well calibrated probability estimates. Inference in multi-class classification for GPs is in general a hard task, involving multiple intractabilities (matrix factorisations). One of the attractions of using GPs for classification is that they aught to be well calibrated, which is not always the case for non-probabilistic classifiers. The paper here proceeds in two directions: how to make an efficient multi-class GP (with comparisons to existing methods); and analysis of the calibration properties of such methods. GP regression is a much simpler task than GP classification (due to the Gaussian likelihood). This immediately suggest that perhaps by treating the (binary) class labels as regression targets, and ignoring the fact that they are discrete variables (this can be extended to multiple classes through a 1vs rest approach). Predictions from such a model may lie outside of the [0, 1] range, so one can either clip the outputs or apply post-calibration. This forms a baseline in the paper. Another idea is to use separate regression models for each class, and then combine these through the use of a categorical/Dirichlet model. I note that this idea is not new - it has been used extensively in the context of Bayesian linear models, and I found several references to this kind of model for GPs [1-4]. The idea of constructing these as separate Gamma likelihoods is novel as far as I am aware, and the neat thing here is that the Gamma likelihood (which would be intractable) can be quite well approximated by the lognormal distribution, allowing inference on log scaled versions of the targets. Note here that the model is heteroskedastic, although there are only two possible values for the noise per instance ($\alpha_i$), which seems somewhat limiting. The analysis of the behaviour of $\alpha$ (section 4.3), shows that it it’s well behaved in testing scenarios. I would conjecture here that the between class variance must be similar in train and test for this to be the case. The experiments are appropriate, and Figure 4 shows that the method is competitive (although not obviously superior) to the competing methods. When combined with the speedup, the comparison with GPC is certainly favourable. It is perhaps harder to argue that it dominates GPR (Platt) and NKRR, although on balance the uplift in performance is probably enough to accept the additional complexity in certain situations. In sum, on the positive side this paper is well written, and contains a (fairly) simple idea that is well executed. The negative side is that there is limited depth to the analysis. Specific comments: 51 “It is established that binary classifiers are calibrated when they employ the logistic loss”. This is under the (strong) assumption that the true conditional distributions of the scores given the class label are Gaussian with equal variance. This of course is often violated in practice. 126 “The Bayes’ rule minimising the expected least squares is the regression function … which in binary classification is proportional to the conditional probability of the two classes”. I guess you mean “The Bayes optimal decision function”? And do you mean the ratio of the conditional probabilities? As in the point above, this being calibrated relies on a conditionally Gaussian assumption 132 breifly -> briefly Figure 1 - are these in fact two-class problems? Table 1 - how are the numbers of inducing points chosen? There’s no obvious pattern Appendix Figure 1 - HTRU2 - GRP (Platt) - I believe this is a case where Platt scaling cannot help [1] Cho, Wanhyun, et al. "Multinomial Dirichlet Gaussian Process Model for Classification of Multidimensional Data." World Academy of Science, Engineering and Technology, International Journal of Computer, Electrical, Automation, Control and Information Engineering9.12 (2015): 2453-2457. [2] Chan, Antoni B. "Multivariate generalized Gaussian process models." arXiv preprint arXiv:1311.0360 (2013). [3] Cho, Wanhyun, et al. "Variational Bayesian Inference for Multinomial Dirichlet Gaussian Process Classification Model." Advances in Computer Science and Ubiquitous Computing. Springer, Singapore, 2015. 835- [4] Salimbeni & Deisenroth. Gaussian process multiclass classification with Dirichlet priors for imbalanced data. Workshop on Practical Bayesian Nonparametrics, NIPS 2016. Note: Following author feedback my concerns about related work have been addressed. I feel that this is a solid if not remarkable paper.